Subject Areas:
biomaterials/biochemistry

Keywords:
poly(lactic-co-glycolic acid) nanoparticle, Box–Behnken design, astaxanthin, anti-photodamage effect

Authors for correspondence:
Fanhui Kong
e-mail: kong_fanhui@foxmail.com
Kun Wei
e-mail: weikun@scut.edu.cn

# Optimization and characterization of poly(lactic-co-glycolic acid) nanoparticles loaded with astaxanthin and evaluation of anti-photodamage effect *in vitro*

Fangbin Hu, Weikang Liu, Liuliu Yan, Fanhui Kong and Kun Wei

School of Bioscience and Bioengineering, South China University of Technology, Guangzhou 510006, People's Republic of China

FH, 0000-0002-2048-3981; LY, 0000-0002-4640-1016

Astaxanthin is a xanthophyll carotenoid with high beneficial biological activities, such as antioxidant function and scavenging oxygen free radicals, but its application is limited because of poor water solubility and low bioavailability. Here, we prepared and optimized poly(lactic-co-glycolic acid) (PLGA) nanoparticles loaded with astaxanthin using the emulsion solvent evaporation technique and investigated the anti-photodamage effect in HaCaT cells. The four-factor three-stage Box–Behnken design was used to optimize the nanoparticle formulation. The experimental determination of the optimal nanoparticle size was $154.4 \pm 0.35$ nm, the zeta potential was $22.07 \pm 0.93$ mV, encapsulation efficiency was $96.42 \pm 0.73\%$ and drug loading capacity was $7.19 \pm 0.12\%$. The physico-chemical properties of the optimized nanoparticles were characterized by dynamic light scattering, scanning electron microscopy, transmission electron microscopy, Fourier-transform infrared spectroscopy, X-ray diffraction, differential scanning calorimetry and thermo-gravimetric analyser. *In vitro* study exhibited the excellent cell viability and cellular uptake of optimized nanoparticles on HaCaT cells. The anti-photodamage studies (cytotoxicity assay, reactive oxygen species content and JC-1 assessment) demonstrated that the optimized nanoparticles were more effective and safer than pure astaxanthin in HaCaT cells. These results suggest that our

PLGA-coated astaxanthin nanoparticles synthesis method was highly feasible and can be used in cosmetics or the treatment of skin diseases.

## 1. Introduction

Astaxanthin is a xanthophyll carotenoid, which widely presents in microorganisms including *Haematococcus pluvialis* and *Phaffia rhodozyma*, or in aquatic animals, such as crabs, salmon and lobsters [1,2]. Furthermore, owing to its numerous biological activities, such as anti-cancer [3], anti-oxidation [4], anti-inflammation [5], decreasing the risk of cardiovascular diseases [6] and photodamage [7], astaxanthin is often applied as a nutritional supplement. Because of its special chemical and molecular structure, the antioxidant activity of astaxanthin is much higher than vitamin E and β-carotene [8]. However, the applications of astaxanthin are a hindrance owing to its high hydrophobicity and poor chemical stability, which also limit its bioavailability [9]. Meanwhile, because it contains a highly unsaturated molecular structure, astaxanthin is sensitive to light, oxide, heat, alkaline and acidic solution. Hence, the use of astaxanthin in the food industry and the pharmaceutical product is severely restricted [10]. Given these conditions, some researchers developed effective strategies such as nanoparticulate drug delivery technologies to overcome these problems and enhance the bioavailability of astaxanthin. For example, Liu *et al.* developed a new delivery system, which encapsulated astaxanthin with β-lactoglobulin and chitosan oligosaccharides to improve the stability and antioxidant activity of astaxanthin [11], and Wu *et al*. reported that astaxanthin was successfully incorporated into hyaluronic acid nanoparticles via an electrostatic field system and had a better recovery effect than astaxanthin alone on retrorsine–CCl$_4$-induced liver fibrosis in the rat model [12].

It is well known that polymeric nanoparticles are the most widely used for hydrophobic drug delivery because of biocompatible and biodegradable, especially poly(D,L-lactic-co-glycolic acid) (PLGA) [13]. PLGA is allowed for use in humans by the United States Food and Drug Administration, as it can be hydrolysed to non-toxic biodegradable metabolite monomers like lactic acid, and glycolic acid *in vivo* [14,15]. Recently, drug delivery based on the use of PLGA nanoparticles has been studied for the treatment of skin damage and photodamage. PLGA-tocopheryl polyethylene glycol 1000 succinate (TPGS) nanoparticles were used to enhance the hydrophilicity of quercetin and could penetrate through the epidermis and reach the dermis to ameliorate in the ultraviolet B (UVB) damaged skin [16]. To reduce the phototoxicity of curcumin (Cur), Deepti *et al*. prepared PLGA-curcumin nanoparticles which could protect the undesirable biological interactions of curcumin photodegradation products under ultraviolet A and UVB exposure and continuously scavenge free radicals [17]. Furthermore, it has been reported that astaxanthin was encapsulated in PLGA nanoparticles coated with chitosan oligosaccharides using an antisolvent precipitation method which could improve water solubility and stability, but the effect of intracellular uptake of nanoparticles has not been proved [18].

The response surface method (RSM), which can be based on experimental results to fit empirical models, is an effective method to optimize the preparation process using a minimum number of experiments, but the results are scientific and credible [19]. Box–Behnken design (BBD), is a kind of RSM, which is a formidable and valid statistical tool for analysing the effects of factors on the response values in pharmaceutical formulation development. It can also obtain optimal preparation conditions of nanoparticles and get a prediction of the response value [20].

Skin, the largest organ in the human body, plays a major role as the protective barrier against harmful external agents, such as ultraviolet (UV) radiation, dehydration, temperature changes and pathogens [21]. Excessive exposure to UV radiation remains a major risk factor for melanoma and non-melanoma skin cancers, especially, exposure to UVB radiation can generate excessive reactive oxygen species (ROS) in cells, that can induce many deleterious effects, including DNA damage, oxidative stress, photoaging, inflammation and carcinogenesis [22,23]. The anti-wrinkle and anti-oxidation effects of astaxanthin reflect its various health benefits and important nutritional health applications in dermatology [24]. Naoki *et al*. evaluated the effects of astaxanthin on UV-induced skin degradation in 23 healthy Japanese participants and demonstrated the protective and safe nature of astaxanthin [25]. Moreover, Hung *et al*. found that barrier defects caused by UV radiation may increase the skin penetration of polymer nanoparticles [26].

In our study, for the first time, to our knowledge, we synthesized and optimized PLGA nanoparticles (AST-PLGA NP) loaded with astaxanthin (AST) using the emulsion solvent evaporation method. According to the BBD, the optimized nanoparticles possessed the highest encapsulation rate, the increased drug load capacity with smaller nanoparticle size. The various physico-chemical and morphological

**Table 1.** Factors and levels used in the Box–Behnken design.

| factors | | levels | | |
| --- | --- | --- | --- | --- |
| | | low (−1) | centre point (0) | high (+1) |
| $X_A$ | PLGA concentration (mg ml$^{-1}$) | 10 | 15 | 20 |
| $X_B$ | astaxanthin concentration (mg ml$^{-1}$) | 0.5 | 0.75 | 1 |
| $X_C$ | water volume (ml) | 1 | 2 | 3 |
| $X_D$ | sonication time (min) | 0.5 | 0.75 | 1 |

properties of the optimized AST-PLGA NP were characterized, and the cellular uptake effect of nanoparticles was assessed using HaCaT cells. Furthermore, the *in vitro* photodamage protective effect of optimized AST-PLGA NP and astaxanthin were evaluated.

# 2. Material and methods

## 2.1. Materials

PLGA with terminal carboxylate groups (PLGA-COOH, the molar ratio of D, L-lactic to glycolic acid, 50 : 50, MW 17 kDa) was purchased from Jinan Daigang Biomaterial Co., Ltd (Shandong, China). Astaxanthin (AST, SA8730) was purchased from Solarbio (Beijing, China). Fluorescein isothiocyanate (FITC, F7250–100 mg) and 2,7-Dichlorodihydrofluorescein diacetate (DCFH-DA) were purchased from Sigma-Aldrich (UK). Dulbecco's Modified Eagle's medium (DMEM), 0.25% trypsin–EDTA and fetal bovine serum (FBS) were purchased from GIBCO (Invitrogen Corp, Carlsbad, CA, USA). The Mitochondrial Membrane Potential Detection Kit (JC-1) was obtained from Beyotime Biotechnology (Shanghai, China). All other reagents were purchased from Aladdin (Shanghai, China).

## 2.2. Preparation of poly(lactic-co-glycolic acid) loaded with astaxanthin nanoparticles

The nanoparticles were prepared following the emulsion solvent evaporation technique [27]. Briefly, an accurately weighed amount of PLGA and astaxanthin were dissolved in a 1 ml mixture of dichloromethane and acetone (dichloromethane: acetone = 3 : 2) under probe sonication. This organic phase was slowly added to bovine serum albumin (BSA) 1% (w/v) solution and sonicated together. The final emulsion was maintained under magnetic stirring at room temperature overnight. After centrifuging at 14 000 rpm for 40 min, the PLGA loaded with astaxanthin nanoparticles were washed three times with deionized water. The unloaded nanoparticles (PLGA NP) were obtained following the same procedure without astaxanthin. The optimized preparation method through response surface experiment design is as follows.

## 2.3. Response surface methodology experiment design

In order to explore optimal formulation condition, we employ the RSM to evaluate the relationship between factors and responses using a minimum number of experiments and use BBD to analyse the results [28]. In this study, the experiment was employed with four factors, three levels and 29 runs for the optimization study using DESIGN-EXPERT 8.0.6 software. The concentration of PLGA ($X_A$), the concentration of astaxanthin ($X_B$), water volume ($X_C$) and sonication time ($X_D$) were selected as independent variables and they were set at low, medium and high levels, respectively (table 1). Then, different batches were prepared with different independent variables at different levels, and responses like size, drug loading (DL) capacity and encapsulation efficiency (EE) were obtained. The data were analysed using the DESIGN-EXPERT software, and polynomial model equations were given as follows:

$$Y = \beta0 + \beta1X_A + \beta2X_B + \beta3X_C + \beta4X_D + \beta5X_AX_B + \beta6X_AX_C + \beta7X_AX_D$$
$$+ \beta8X_BX_C + \beta9X_BX_D + \beta10X_CX_D + \beta11X_A^2 + \beta12X_B^2 + \beta13X_C^2 + \beta14X_D^2, \quad (2.1)$$

where $Y$ is the individual response factor or dependent variable; $\beta0$–$\beta11$ are regression coefficients; and $X_A$, $X_B$, $X_C$ and $X_D$ are the independent variables.

## 2.4. Particle size and zeta potential

Particle size, polydispersity index (PDI) and zeta potential of the nanoparticles were analysed by the method of dynamic light scattering (DLS) (Zetasizer Nano ZS, Malvern, UK). Each measurement was repeated three times.

## 2.5. Evaluation of the encapsulation efficiency and drug loading capacity

The EE and DL of AST-PLGA NP were determined by a UV–visible spectrophotometer (UV-2600, Shimadzu, Japan). One milligram lyophilized AST-PLGA NP was accurately weighed, and astaxanthin in AST-PLGA NP was extracted by adding 1 ml dichloromethane: ethanol (1 : 10, v : v). The above extract was vortexed and centrifuged at 10 000 rpm for 5 min and the supernatant collected, which was measured at the wavenumber of 478 nm. The EE and DL were calculated using the following equation:

$$EE\% = \frac{\text{weight of encapsulated drug}}{\text{total weight of drug}} \times 100\% \tag{2.2}$$

$$DL\% = \frac{\text{weight of encapsulated drug}}{\text{weight of the loaded nanoparticles}} \times 100\%. \tag{2.3}$$

## 2.6. Morphologic analysis of nanoparticles

The shape and size of the nanoparticles were observed using a scanning electron microscopic (SEM) (Merlin, Carl Zeiss AG, Germany) at an acceleration voltage of 5 kV. A drop of 1 mg ml$^{-1}$ sample was placed on a wafer, dried overnight and then coated with platinum for the experiment. The surface morphology of nanoparticles was further characterized by transmission electron microscopy (TEM).

## 2.7. Fourier-transform infrared spectroscopy

The Fourier-transform infrared (FTIR) spectra of astaxanthin, free PLGA nanoparticles and AST-PLGA NPs were recorded by a Thermo Scientific Nicolet Nicolet Nexus FTIR spectrometer (Thermo-Nicolet, Gaisburg, MD, USA) in the ranges of 400–4000 cm$^{-1}$ for chemical analysis of functional groups. Samples were ground together with KBr (1 : 50, w/w).

## 2.8. X-ray diffraction patterns

A certain amount of astaxanthin, free PLGA NP and AST-PLGA NP were scanned on a powder X-ray diffractometer (XTV160H, X-TEK, U.K) with an angle of 2$\theta$ from 5 to 60° under ambient temperature.

## 2.9. Thermal analysis

The thermal properties of astaxanthin, PLGA NP and AST-PLGA NP were measured using a differential scanning calorimetry (DSC) and thermo-gravimetric analyser (TGA). Both DSC and TGA profiles were accomplished on an STA 449C (Netzsch, Germany). Each sample (10 mg) was put into an aluminium pan and heated from 35 to 800°C at a flow rate of 10 K min$^{-1}$ under the flow of nitrogen.

## 2.10. Cell culture

Human keratinocytes (HaCaT) were purchased from the Kunming Cell Bank of the Committee of Typical Culture Collection of Chinese Academy of Sciences. HaCaT cells were cultured in DMEM media supplemented with 1% of penicillin/streptomycin and 10% FBS in 5% $CO_2$ at 37°C. When reaching 80% confluence, cells were detached upon trypsinization for experiments.

## 2.11. Cell uptake

Flow cytometry analysis: HaCaT cells were cultured at $1 \times 10^5$ cells well$^{-1}$ in 12 well plates. When the cells properly adhered, the medium was removed and FITC-loaded PLGA nanoparticle suspension (100 µg ml$^{-1}$) was incubated for 1, 2, 4 and 8 h. After different exposure times, cells were washed

twice with phosphate buffered saline (PBS) and harvested using trypsin–EDTA (0.25% v/v). The results were recorded using a BD C6 flow cytometry (FACS) with a 488 nm laser. For each experiment, a total of 10 000 cells were gated per sample and each sample was performed in triplicate.

Confocal laser scanning microscopy: laser confocal microscopy (Ti-E A1, Nikon, Japan) was used to qualitatively analyse the localization of fluorescent nanoparticles in HaCaT cells. The cells were seeded at $5 \times 10^4$ cells well$^{-1}$ in 24-well plates containing glass coverslips. After incubation with fluorescent nanoparticles for a certain period of time, the cells were washed gently with PBS three times and fixed with 4% paraformaldehyde for 20 min. Finally, cells were stained with 4,6-diamidino-2 phenylindole (DAPI) for 5 min.

## 2.12. In *vitro* cell viability and cytotoxicity studies

Cell viability was measured by the colorimetric 3-(4,5-dimethylthiazol-2-yl)-2, 5-diphenyltetrazolium bromide (MTT) method. HaCaT cells were seeded at $8 \times 10^3$ cells well$^{-1}$ in 96-well plates. The different concentrations of astaxanthin, free PLGA NP and AST-PLGA NP suspension were treated for 24 h and 48 h, respectively. Then, the media was removed and replaced by the MTT solution (0.5 mg ml$^{-1}$ well$^{-1}$). After incubation at 37°C for 4 h, 100 µl dimethyl sulfoxide dissolved the formazan crystals, and the absorbance was measured at 570 nm using a microplate reader (Cytation 5, Biotek, USA).

## 2.13. Ultraviolet B irradiation and phototoxicity measurement

Cells were irradiated with UVB from two fluorescent Philips lamps (280–370 nm) with a peak at 312 nm and measured by a UVB radiometer with a sensor. Cells were pretreated with or without astaxanthin, and AST-PLGA NP for 12 h, and exposed to the dose of 30 mJ cm$^{-2}$ irradiation. Cells were rinsed by PBS once and irradiated under a thin layer of PBS. After UVB radiation, cells were incubated with the drug or free media for another 24 h immediately. The control group was treated to the same procedures without exposing to UVB lamps. The protection of phototoxicity of astaxanthin and AST-PLGA NP was assessed by the MTT method.

## 2.14. Assessment of intracellular reactive oxygen species

The fluorescence dye 2′,7′-dichlorodihydrofluorescein diacetate (DCFH-DA) (Sigma-Aldrich, USA) was used to investigate intracellular ROS. HaCaT cells were seeded at $1.5 \times 10^5$ cells well$^{-1}$ in 12-well plates overnight. The above irradiation method was carried on until after the cells reached 80% confluence. Finally, the cells were incubated with 10 µM DCFH-DA in serum-free medium for 30 min, and the results were analysed by FACS and confocal laser scanning microscopy.

## 2.15. Measurement of mitochondrial membrane potential

Mitochondrial membrane potential ($\Delta\Psi$m) was measured by a mitochondrial membrane potential assay kit with JC-1 (C2006, Beyotime Biotechnology, Jiangsu, China) according to the manufacturer's kit instructions. In brief, after the irradiation treatment mentioned above, cells were washed with PBS and treated with JC-1 at 37°C for 20 min in dark conditions. Then the results of the fluorescence intensity were recorded by FACS and confocal laser scanning microscopy. The value of mitochondrial membrane potential was calculated using the following equation:

$$\Delta\psi\mathrm{m} = \frac{\mathrm{red\,(J-aggregates)}}{\mathrm{green\,(JC-imonomer)}} \times 100\%. \tag{2.4}$$

## 2.16. Statistical analysis

The experiments were repeated three times separately and the results are presented as the means ± s.e.m. The statistical significance of the results was evaluated using an independent *t*-test for the comparison of two samples. A *p*-value of less than 0.05 was considered statistically significant.

**Table 2.** Box–Behnken design matrix and observed response value.

| input factor levels | | | | | experimental response | | |
| --- | --- | --- | --- | --- | --- | --- | --- |
| run | $X_A$ | $X_B$ | $X_C$ | $X_D$ | EE (%) | DL (%) | size (nm) |
| 1 | 20 | 0.75 | 1 | 0.75 | 97.45 | 3.58 | 261.43 |
| 2 | 15 | 0.75 | 1 | 1 | 93.30 | 4.70 | 350.23 |
| 3 | 10 | 0.75 | 2 | 1 | 94.08 | 6.56 | 179.03 |
| 4 | 15 | 0.5 | 3 | 0.75 | 83.00 | 2.46 | 148.17 |
| 5 | 20 | 0.75 | 3 | 0.75 | 90.60 | 3.27 | 158.80 |
| 6 | 10 | 0.75 | 2 | 0.5 | 95.09 | 6.63 | 193.47 |
| 7 | 15 | 0.75 | 2 | 0.75 | 84.31 | 4.01 | 192.00 |
| 8 | 20 | 0.5 | 2 | 0.75 | 92.60 | 2.26 | 225.00 |
| 9 | 15 | 0.75 | 2 | 0.75 | 89.48 | 4.26 | 171.83 |
| 10 | 15 | 0.5 | 2 | 1 | 93.78 | 3.03 | 181.90 |
| 11 | 20 | 0.75 | 2 | 0.5 | 93.07 | 3.08 | 238.77 |
| 12 | 15 | 0.75 | 2 | 0.75 | 81.95 | 3.80 | 189.40 |
| 13 | 10 | 0.75 | 1 | 0.75 | 88.02 | 6.14 | 369.27 |
| 14 | 15 | 1 | 2 | 0.5 | 82.61 | 5.16 | 200.03 |
| 15 | 15 | 0.5 | 1 | 0.75 | 85.52 | 2.76 | 265.87 |
| 16 | 15 | 1 | 1 | 0.75 | 77.81 | 4.86 | 267.57 |
| 17 | 10 | 0.75 | 3 | 0.75 | 91.72 | 6.40 | 137.10 |
| 18 | 20 | 0.75 | 2 | 1 | 95.09 | 3.44 | 198.53 |
| 19 | 15 | 1 | 3 | 0.75 | 77.81 | 4.86 | 153.00 |
| 20 | 15 | 0.75 | 2 | 0.75 | 80.27 | 3.62 | 194.93 |
| 21 | 15 | 0.75 | 3 | 1 | 90.49 | 4.31 | 166.97 |
| 22 | 15 | 0.75 | 1 | 0.5 | 92.96 | 4.43 | 266.27 |
| 23 | 10 | 0.5 | 2 | 0.75 | 95.30 | 4.54 | 172.00 |
| 24 | 15 | 0.75 | 3 | 0.5 | 91.72 | 4.37 | 165.10 |
| 25 | 10 | 1 | 2 | 0.75 | 82.61 | 7.51 | 148.97 |
| 26 | 15 | 1 | 2 | 1 | 75.03 | 4.69 | 146.40 |
| 27 | 20 | 1 | 2 | 0.75 | 86.74 | 4.13 | 165.03 |
| 28 | 15 | 0.5 | 2 | 0.5 | 91.25 | 2.94 | 150.50 |
| 29 | 15 | 0.75 | 2 | 0.75 | 84.53 | 4.03 | 169.10 |

# 3. Results and discussion

## 3.1. Formulation and optimization of astaxanthin-loaded poly(lactic-co-glycolic acid) nanoparticles by Box–Behnken design

BBD can study the effects and interactions between all factors and responses with the least number of experiments [29]. In our study, the four-factor, three-level experiment design was analysed with three responses to explore the best formulation. The higher and lower limits were chosen in pilot studies. The results of the whole design contained 29 experiment runs with five central points, displayed in table 2 in the form of means ($n = 3$). The EE, DL and size of the AST-PLGA NP ranged from 75.03–97.45%, 2.26–7.51% and 137.1–369.27 nm, respectively. Various model statistical parameters were analysed by DESIGN-EXPERT 8.0.6 software using multiple regression analysis.

**Table 3.** Statistical analysis of variance for EE in Box–Behnken design. ($R^2 = 0.8772$; $R^2_{Adj} = 0.75450$; CV = 3.45%.)

| source | sum of squares | mean square | $F$-value | $p$-value |
|---|---|---|---|---|
| model | 924.83 | 66.06 | 7.15 | 0.0004 significant |
| $X_A$ | 6.36 | 6.36 | 0.69 | 0.4209 |
| $X_B$ | 288.63 | 288.63 | 31.22 | <0.0001 |
| $X_C$ | 7.87 | 7.87 | 0.85 | 0.37 |
| $X_D$ | 2.04 | 2.04 | 0.22 | 0.65 |
| $X_{AB}$ | 11.64 | 11.64 | 1.26 | 0.28 |
| $X_{AC}$ | 27.87 | 27.87 | 3.01 | 0.10 |
| $X_{AD}$ | 2.30 | 2.30 | 0.25 | 0.63 |
| $X_{BC}$ | 1.60 | 1.60 | 0.17 | 0.68 |
| $X_{BD}$ | 25.55 | 25.55 | 2.76 | 0.12 |
| $X_{CD}$ | 0.62 | 0.62 | 0.07 | 0.80 |
| $X_A^2$ | 288.95 | 288.95 | 31.25 | <0.0001 |
| $X_B^2$ | 63.04 | 63.04 | 6.82 | 0.02 |
| $X_C^2$ | 13.20 | 13.20 | 1.43 | 0.25 |
| $X_D^2$ | 158.09 | 158.09 | 17.10 | 0.001 |
| residual | 129.44 | 9.25 | | |
| lack of fit | 80.98 | 8.10 | 0.67 | 0.72 not significant |
| pure error | 48.46 | 12.11 | | |
| corrected total | 1054.26 | | | |

### 3.1.1. Effect of formulation variable on encapsulation efficiency

We used the quadratic model to analyse the variance of the BBD experiment. ANOVA results for EE response are shown in table 3. The full regression model was significant ($p = 0.0004$) with insignificant lack of fit ($F = 0.67$; $p = 0.72$), which meant that it was excellent fitting to the corresponding response and a quadratic second-order polynomial model was fitted with the equation as follows:

$$EE(Y_2) = 84.11 + 0.73X_A - 4.9X_B - 0.81X_C - 0.41X_D + 1.71X_AX_B - 2.64X_AX_C + 0.76X_AX_D$$
$$+ 0.63X_BX_C - 2.53X_BX_D - 0.39X_CX_D + 6.67X_A^2 - 3.12X_B^2 + 1.43X_C^2 + 4.94X_D^2. \tag{3.1}$$

From the equation, a positive value indicated a synergistic effect on optimization in the regression model, whereas a negative value indicated an antagonistic effect [30]. Factors A, AB, AD, BC, $A^2$, $C^2$ and $D^2$ had a positive effect on the EE. However, factors B, C, D, AC, BD, CD and $B^2$ showed passive influence on EE. In table 3, the $R^2$ values and the adjusted $R^2$ values were 0.8772 and 0.7545, respectively, and the value of the coefficient of variation (CV) was 3.45%, which suggested that the fitted model was in reasonable agreement.

### 3.1.2. Effect of formulation variable on drug loading

High DL capacity in nanoparticles can decrease drug dosing frequency and side effects [31]. ANOVA results of DL are shown in table 4. The fitted model was extremely significant ($F = 61.47$ and $p < 0.0001$). The lack of fit was not significant (0.6) and a quadratic second-order polynomial model was fitted with the equation as follows:

$$DL(Y_2) = 3.94 - 1.5X_A + 1.1X_B - 0.066X_C + 0.009X_D - 0.28X_AX_B - 0.14X_AX_C + 0.11X_AX_D$$
$$+ 0.075X_BX_C - 0.14X_BX_D - 0.083X_CX_D + 0.8X_A^2 - 0.24X_B^2 + 0.12X_C^2 + 0.27X_D^2. \tag{3.2}$$

In the model, the $R^2$ values and the adjusted $R^2$ values were 0.984 and 0.968, respectively. The value was close to 1 indicating a good fit in reasonable agreement. Among the factors, B, D, AD, BC, $A^2$, $C^2$ and

**Table 4.** Statistical analysis of variance for DL in Box–Behnken design. ($R^2 = 0.9840$; $R^2_{Adj} = 0.9680$; CV = 5.43%.)

| source | sum of squares | mean square | F-value | p-value |
|---|---|---|---|---|
| model | $4.77 \times 10^{-3}$ | $3.41 \times 10^{-4}$ | 61.47 | <0.0001 significant |
| $X_A$ | $2.71 \times 10^{-3}$ | $2.71 \times 10^{-3}$ | 488.33 | <0.0001 |
| $X_B$ | $1.46 \times 10^{-3}$ | $1.46 \times 10^{-3}$ | 263.21 | <0.0001 |
| $X_C$ | $5.20 \times 10^{-6}$ | $5.20 \times 10^{-6}$ | 0.94 | 0.35 |
| $X_D$ | $9.06 \times 10^{-8}$ | $9.06 \times 10^{-8}$ | 0.02 | 0.90 |
| $X_{AB}$ | $3.03 \times 10^{-5}$ | $3.03 \times 10^{-5}$ | 5.46 | 0.03 |
| $X_{AC}$ | $7.81 \times 10^{-6}$ | $7.81 \times 10^{-6}$ | 1.41 | 0.26 |
| $X_{AD}$ | $4.49 \times 10^{-6}$ | $4.49 \times 10^{-6}$ | 0.81 | 0.38 |
| $X_{BC}$ | $2.23 \times 10^{-6}$ | $2.23 \times 10^{-6}$ | 0.40 | 0.54 |
| $X_{BD}$ | $7.71 \times 10^{-6}$ | $7.71 \times 10^{-6}$ | 1.39 | 0.26 |
| $X_{CD}$ | $2.75 \times 10^{-6}$ | $2.75 \times 10^{-6}$ | 0.50 | 0.49 |
| $X_A^2$ | $4.15 \times 10^{-4}$ | $4.15 \times 10^{-4}$ | 74.91 | <0.0001 |
| $X_B^2$ | $3.82 \times 10^{-5}$ | $3.82 \times 10^{-5}$ | 6.89 | 0.02 |
| $X_C^2$ | $1.00 \times 10^{-5}$ | $1.00 \times 10^{-5}$ | 1.80 | 0.20 |
| $X_D^2$ | $4.90 \times 10^{-5}$ | $4.90 \times 10^{-3}$ | 8.84 | 0.01 |
| residual | $7.76 \times 10^{-5}$ | $5.55 \times 10^{-6}$ | | |
| lack of fit | $5.37 \times 10^{-5}$ | $5.37 \times 10^{-6}$ | 0.90 | 0.60 not significant |
| pure error | $2.40 \times 10^{-5}$ | $5.99 \times 10^{-6}$ | | |
| corrected total | $4.85 \times 10^{-3}$ | | | |

$D^2$ had a positive effect on the size of nanoparticles and the concentration of PLGA (A) and astaxanthin (B) had the highest significant effect on the size with $p < 0.0001$, indicating that A and B are the most important factors for DL.

### 3.1.3. Effect of formulation variable on size

It is well known that the size and PDI of nanoparticles are very important features, which may influence toxicity, stability and biodistribution of drug delivery [32]. The size of nanoparticles was seen as a response, ANOVA results are shown in table 5. The fitted model was highly significant ($p = 0.0005$ and $F = 6.77$). Meanwhile, the lack of fit was not significant which means it is a good fit to the corresponding response, and a quadratic second-order polynomial model was fitted with the equation as follows:

$$\text{Size}(Y_3) = 183.93 + 3.98X_A - 5.2X_B - 70.97X_C + 0.73X_D - 9.230X_AX_B + 32.38X_AX_C - 6.45X_AX_D$$
$$+ 0.78X_BX_C - 21.26X_BX_D - 20.48X_CX_D + 9.4X_A^2 - 18.47X_B^2 + 42.21X_C^2 + 8.14X_D^2. \tag{3.3}$$

In the model, the $R^2$ values and the adjusted $R^2$ values were 0.8713 and 0.7425, respectively, in reasonable agreement. Among the factors, A, D, AC, BC, $A^2$, $C^2$ and $D^2$ had a positive effect on the size of nanoparticles, and water volume (C) had the highest significant effect on the size with $p < 0.0001$. The change of water volume actually changed the ratio of the W/O phase. When the surfactant (BSA) concentration remained unchanged and increased the water concentration, the particle size decreased. These results were consistent with the previous study [28].

### 3.1.4. Optimization and model validation

The optimized PLGA NP loaded with astaxanthin formulation was performed based on the maximum value of EE and DL, meanwhile minimizing the particle size by using the prediction method of Design-Expert 8.0.6 software. The optimized formation composition is shown in table 6. PLGA NP

**Table 5.** Statistical analysis of variance for size in Box–Behnken design. ($R^2 = 0.8713$; $R^2_{Adj} = 0.7425$; CV = 14.97%.)

| source | sum of squares | mean square | F-value | p-value |
|---|---|---|---|---|
| model | 85 733.94 | 6123.85 | 6.77 | 0.0005 significant |
| $X_A$ | 189.87 | 189.87 | 0.21 | 0.65 |
| $X_B$ | 324.83 | 324.83 | 0.36 | 0.56 |
| $X_C$ | 60 444.68 | 60 444.68 | 66.80 | <0.0001 |
| $X_D$ | 6.40 | 6.40 | 0.01 | 0.93 |
| $X_{AB}$ | 341.02 | 341.02 | 0.38 | 0.55 |
| $X_{AC}$ | 4194.72 | 4194.72 | 4.64 | 0.05 |
| $X_{AD}$ | 166.41 | 166.41 | 0.18 | 0.67 |
| $X_{BC}$ | 2.45 | 2.45 | 0.00 | 0.96 |
| $X_{BD}$ | 1807.67 | 1807.67 | 2.00 | 0.18 |
| $X_{CD}$ | 1678.27 | 1678.27 | 1.85 | 0.19 |
| $X_A^2$ | 572.71 | 572.71 | 0.63 | 0.44 |
| $X_B^2$ | 2213.87 | 2213.87 | 2.45 | 0.14 |
| $X_C^2$ | 11 558.54 | 11 558.54 | 12.77 | 0.00 |
| $X_D^2$ | 430.03 | 430.03 | 0.48 | 0.50 |
| residual | 12 668.27 | 904.88 | | |
| lack of fit | 9573.73 | 957.37 | 1.24 | 0.45 not significant |
| pure error | 3094.54 | 773.64 | | |
| corrected total | 98 402.21 | | | |

**Table 6.** Optimized values obtained by constraints applied on EE, DL, size.

| variable and response | optimum condition | | | | predicted value | observed value | bias (%) |
|---|---|---|---|---|---|---|---|
| | $X_A$ | $X_B$ | $X_C$ | $X_D$ | | | |
| EE | 10 | 0.81 | 3 | 0.5 | 98.85 | 96.42 | 2.46 |
| DL | 10 | 0.81 | 3 | 0.5 | 7.26 | 7.19 | 0.96 |
| size | 10 | 0.81 | 3 | 0.5 | 154.68 | 154.4 | 0.18 |

bias = (predicted value − observed value)/predicted value × 100%

loaded with astaxanthin was synthesized at the optimal conditions three times to verify. The experimentally observed values for the three responses are close to the predicted values with low percentages of the relative error to ensure the validity and reproducibility of the model [33]. Figure 1 shows the linear relationship between the experimental value of the response and the predicted value. The more uniformly the point is close to the 45° line, the better the model fits [34]. The synthesised method of optimized PLGA NP loaded with astaxanthin was prepared in the remaining experiments.

## 3.2. Characterization of the optimized poly(lactic-co-glycolic acid) nanoparticles loaded with astaxanthin

The mean particle size of optimized AST-PLGA NP was 154.4 ± 0.35 nm, and the PDI was 0.16 ± 0.032, while the PLGA NP without astaxanthin using the same method was 147.5 ± 1.48 nm and 0.256 ± 0.012 with narrow size distribution, respectively (figure 2a). The size of AST-PLGA NP was slightly larger

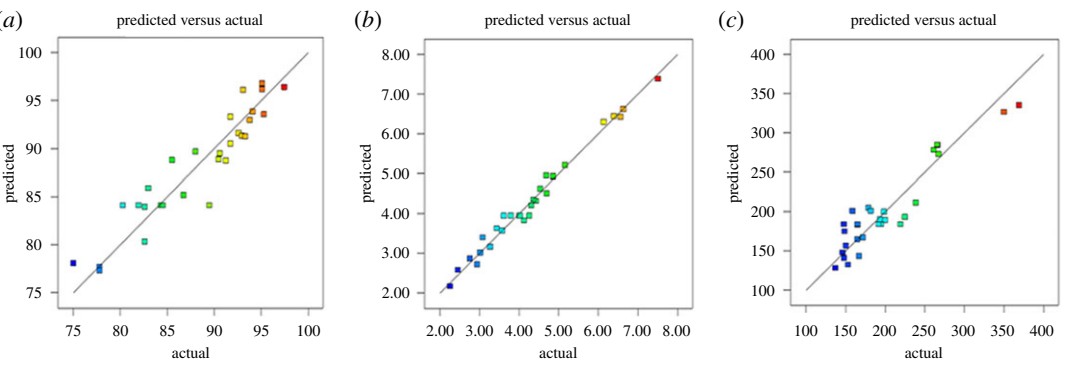

**Figure 1.** Linear correlation plots between actual and predicted values and the corresponding residual plots for (*a*) EE; (*b*) DL and (*c*) size.

**Figure 2.** (*a*) The particle size distribution of PLGA NP and AST-PLGA NP. Scanning electron microscopic (SEM) images of (*b*) astaxanthin (AST), scale bar, 10 µm; (*c*) blank PLGA nanoparticles (PLGA NP); and (*d*) astaxanthin-loaded PLGA nanoparticles (AST-PLGA NP). Transmission electron microscopy (TEM) images of (*e*) PLGA NP, and (*f*) AST-PLGA NP, scale bar, 100 nm.

than blank PLGA NP, indicating that astaxanthin successfully loaded into PLGA NP. Furthermore, PLGA NP and AST-PLGA NP showed negative zeta potential, which was $-25.93 \pm 0.58$ mV and $-22.07 \pm 0.933$ mV, respectively. In addition, the evaluation of the EE and DL capacity of optimized

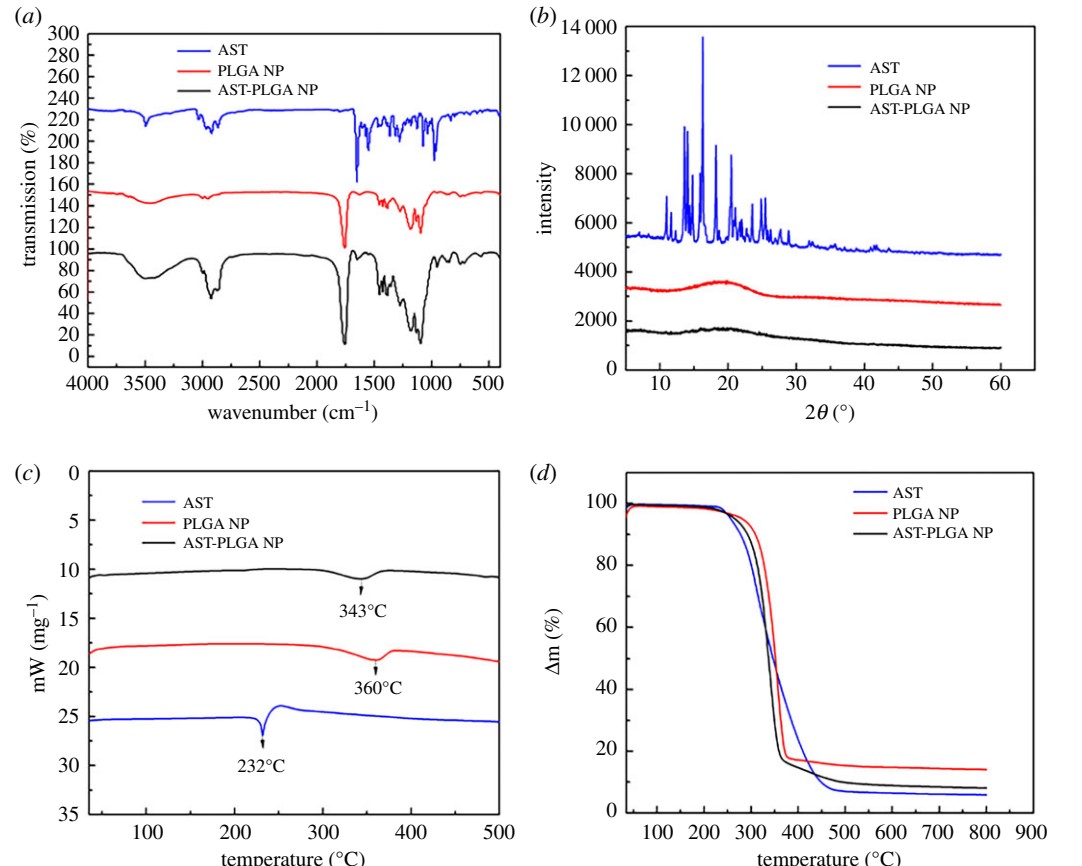

**Figure 3.** Detailed characterization of nanoparticles. (*a*) The FTIR spectrum of astaxanthin, PLGA NP, AST-PLGA NP. (*b*) X-ray diffraction patterns of astaxanthin, PLGA NP, AST-PLGA NP. (*c*) DSC thermographs curves of astaxanthin, PLGA NP, AST-PLGA NP. (*d*) TGA thermographs curves of astaxanthin, PLGA NP, AST-PLGA NP.

AST-PLGA NP were $96.42 \pm 0.73\%$ and $7.19 \pm 0.12\%$, respectively. It is reported that the EE and DL are the significant parameters because they are directly related to the administration of the nanoparticle's quantity [35]. The EE and DL results suggest that the properties of the optimized AST-PLGA NP was excellent.

SEM micrographs of astaxanthin, blank PLGA NP and AST-PLGA NP are shown in figure 2*b–d*, respectively. It could be seen that pure astaxanthin was an irregularly lumpy crystal. After synthesizing, nanoparticles using the optimized method, PLGA NP and AST-PLGA NP showed small, smooth and globular particles without drug crystal presentation, which suggested that they can more easily penetrate into cells than pure astaxanthin. Besides, the results of TEM further validate the morphology of the nanoparticles.

## 3.3. Fourier-transform infrared analysis of the optimized astaxanthin-loaded poly(lactic-co-glycolic acid) nanoparticles

To investigate whether astaxanthin was successfully loaded in PLGA nanoparticles, the FTIR experiment was carried out. As shown in figure 3, the astaxanthin presented characteristic absorption peaks at $2922 \, \text{cm}^{-1}$, $1650 \, \text{cm}^{-1}$, $1551 \, \text{cm}^{-1}$ and $976 \, \text{cm}^{-1}$, which are attributed to $-CH_3$ stretching, the $C=O$ stretching vibration, the stretching vibration of $C=C$ in the aromatic ring and C–H in the C and C conjugate system, respectively [36,37]. All bands of blank PLGA NP could be found in the spectra of AST-PLGA NP. Compared with the spectra of AST-PLGA NP and astaxanthin, it was obvious that specific functional groups of astaxanthin also appeared in AST-PLGA NP. The results demonstrated that astaxanthin was successfully encapsulated into PLGA nanoparticles physically.

## 3.4. X-ray diffraction analysis of the optimized astaxanthin-poly(lactic-co-glycolic acid) nanoparticles

As shown in figure 3b, the characteristic crystalline peaks of astaxanthin occurred at 10.7°, 14.2°, 16.4°, 18.2°, 20.4° and 24.7°, respectively, which meant that the astaxanthin still maintained the crystalline form. Besides, the blank PLGA NP and AST-PLGA NP showed a wider diffraction peak around 11.4° to 25.9°, which is similar to a previous report [38]. Otherwise, as for AST-PLGA NP, the sharpness peaks of astaxanthin disappeared in the X-ray diffraction (XRD) spectra, which suggested that astaxanthin was successfully encapsulated in an amorphous form. These results are similar to a previous report [18].

## 3.5. Thermal analysis of the optimized astaxanthin-loaded poly(lactic-co-glycolic acid) nanoparticles

The results of DSC thermograms are shown in figure 3c. The endothermic peaks of astaxanthin, pure PLGA NP and AST-PLGA NP are observed clearly. The curves of astaxanthin (figure 3c, blue line) showed a sharp endothermic peak at 230°C [39]. However, AST-PLGA NP (figure 3c, red line) only displays a sharp endothermic peak around 350°C like pure PLGA NP (figure 3c, black line) [40]. The disappearance of the astaxanthin characteristic peaks in AST-PLGA NP proved that astaxanthin was successfully wrapped into polymer nanoparticles in an amorphous state.

Figure 3d shows the TGA curves of astaxanthin, blank PLGA NP and AST-PLGA NP respectively. There were two stages of weight loss. The weight loss of the first stage up to 100°C may be the physical adsorption by water in those test compounds [41]. The mass loss of astaxanthin and AST-PLGA NP at 256°C were around 4.8% and 3.9%, respectively, while about 93.3% and 90.7% was a loss at 550°C, respectively. In other words, these data further indicated the presence of astaxanthin in the PLGA nanoparticles [42].

## 3.6. Cellular uptake study

To demonstrate that PLGA nanoparticles were uptake by HaCaT cells, FITC was chosen as a model fluorescent probe to perform a quantitative and qualitative analysis of cellular uptake of the nanoparticles [43]. The results of FACS could quantitatively analyse the uptake of fluorescent nanoparticles by HaCaT cells as shown in figure 4a,b. The cellular uptakes of FITC-loaded PLGA nanoparticles at a concentration of 100 µg ml$^{-1}$ were time-dependent. After incubation with cells for 0 h, 1 h, 2 h, 4 h and 8 h, the fluorescence intensity of the fluorescent nanoparticles were $2214 \pm 48.31$; $5989 \pm 65.02$; $7521 \pm 39.01$; $10\,007 \pm 76.43$; $18\,758 \pm 76.41$, respectively. The intracellular uptake of FITC-loaded PLGA nanoparticles in HaCaT cells was visualized using a laser scanning confocal microscope. FITC-loaded PLGA nanoparticles (green) were distributed around the nucleus (blue) which proved that nanoparticles can be taken up by the cells through the cell membrane (figure 4c). After fluorescent nanoparticles were treated with HaCaT cells for 1 h, the green fluorescence intensity was very weak. However, the green fluorescence intensity was obvious at 8 h. These results confirmed that the cellular uptake of PLGA nanoparticles increased in a time-dependent way.

## 3.7. Cell viability and photoprotective studies

The *in vitro* cytotoxicity of astaxanthin, blank PLGA NP and AST-PLGA NP in HaCaT cells were evaluated using MTT assay (figure 5a,b). The HaCaT cells treated with astaxanthin did not cause any significant decrease in cell viability, while at 48 h, cell viability of 2, 4 and 8 µM was less than 95%. So, we chose 0.4, 0.8 and 1 µM for later experiments. As previously reported [18], the biocompatibility of PLGA NP and AST-PLGA NP had lower cytotoxicity at high concentration at 24 and 48 h.

As shown in figure 5c,d, the photo protective effects of astaxanthin and AST-PLGA NP on HaCaT cells under UVB irradiation of 30 mJ cm$^{-2}$ were evident. The results displayed that the cell viability of irradiated cells was significantly reduced by 42.53% ($p < 0.005$) compared with control cells, which means that the irradiation model was appropriate [44]. The data showed astaxanthin at 1 µM and 1 µM AST-PLGA NP increased cell viability about 16.10% and 18.14%, respectively, indicating that AST-PLGA NP possessed increased UV protection compared to free astaxanthin.

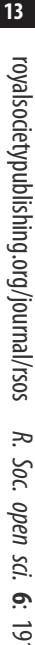

**Figure 4.** Cell uptake study. (*a*) Fluorescence intensity of FITC loaded into PLGA NP in HaCaT cells with different incubation time. The data show the means ± s.e.m. (*n* = 3), ****$p < 0.0001$. (*b*) Flow cytometry results of HaCaT cells cultured with FITC-loaded nanoparticles at different time points. (*c*) The fluorescent images of HaCaT cells treated with FITC-loaded PLGA NP for certain different times using a laser scanning confocal microscope. Scale bar, 50 µm.

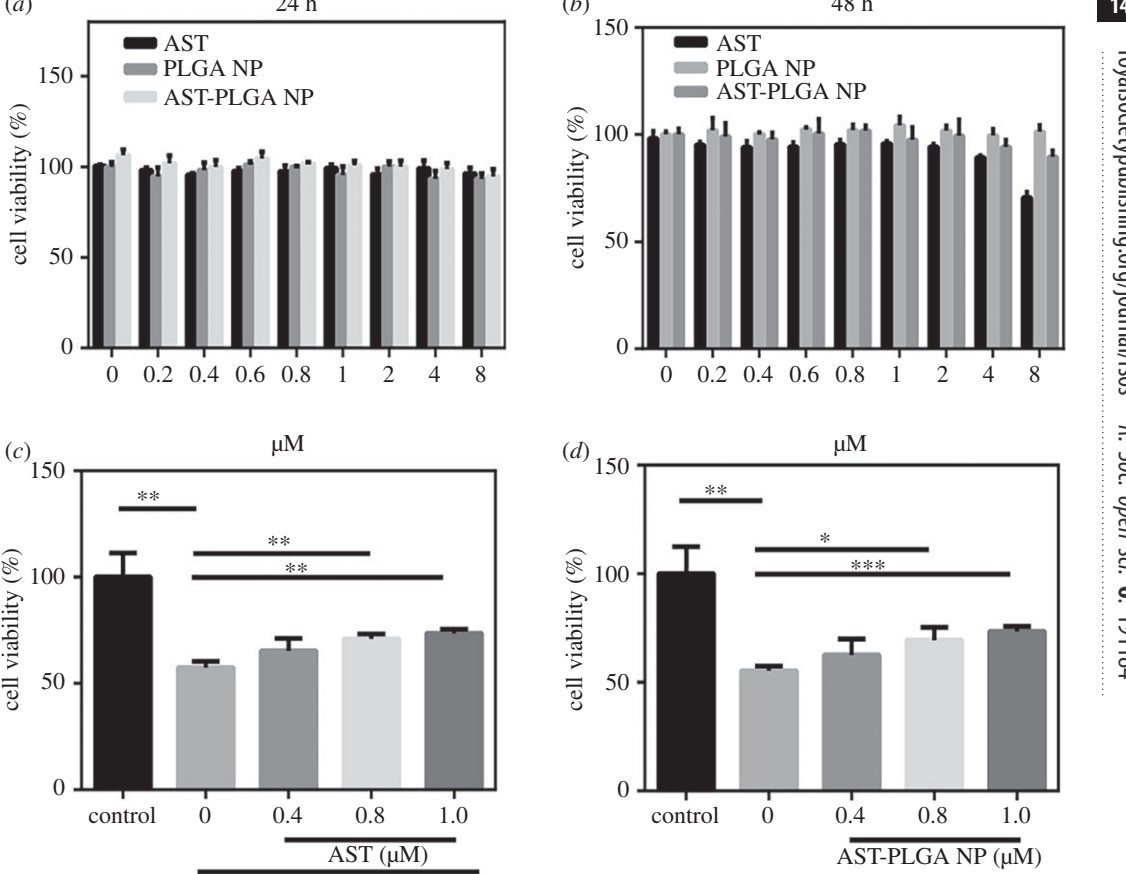

**Figure 5.** Cell viability of astaxanthin (AST) and astaxanthin-loaded PLGA nanoparticles (AST-PLGA NP) treatment and UVB irradiation on HaCaT cells. (*a*) HaCaT cells were treated with indicated concentrations of astaxanthin, blank PLGA NP and AST-PLGA NPs for 24 h. (*b*) HaCaT cells were treated with indicated concentrations of astaxanthin, blank PLGA NP and AST-PLGA NP for 48 h. (*c*) HaCaT cells pretreated with 0.4, 0.8 and 1 μM astaxanthin for 12 h were irradiated with 30 mJ cm$^{-2}$ of UVB for a further 24 h. (*d*) HaCaT cells pretreated with AST-PLGA NP loaded astaxanthin (0.4, 0.8 and 1 μM) for 12 h were irradiated with 30 mJ cm$^{-2}$ of UVB for a further 24 h. The cell viability was evaluated by MTT assay. The results are presented as the mean ± s.e.m. ($n = 3$). *$p < 0.05$, **$p < 0.001$, ***$p < 0.005$, ****$p < 0.0001$.

## 3.8. Measurement of reactive oxygen species generation

UVB radiation can cause a mass of ROS, which induces oxidative stress and consequently accounts for cellular compensation which ultimately leads to programmed cell death [45]. In our study, ROS levels in HaCaT cells was obviously increased after 30 mJ cm$^{-2}$ of UVB irradiation compared with control cells (by $297.7 \pm 14.46\%$). As shown in figure 6*a*, astaxanthin pre-treatment decreased the intracellular ROS generation and reduced DCF fluorescence intensity; the dose–response effect was observed. The influence on ROS level of AST-PLGA NP had a similar result, which indicated AST-PLGA NP did not change the antioxidant properties of astaxanthin. Furthermore, the results of the laser scanning confocal microscope for astaxanthin and AST-PLGA NP testifies that the ROS level was quenched and varies with concentration. In a word, astaxanthin and AST-PLGA NP can both protect HaCaT cells from UVB-induced oxidative stress by scavenging free radicals.

## 3.9. Measurement of mitochondrial membrane depolarization

Mitochondrial membrane depolarization, which caused ΔΨm to lose, is regarded as a hallmark of the early and irreparable stage of cell apoptosis [46]. Red fluorescence stands for the mitochondrial aggregated form of JC-1 because of a high ΔΨm, while green fluorescence represents the monomeric form of JC-1 owing to low ΔΨm [47]. Variation in ΔΨm leads to mitochondrial dysfunction and

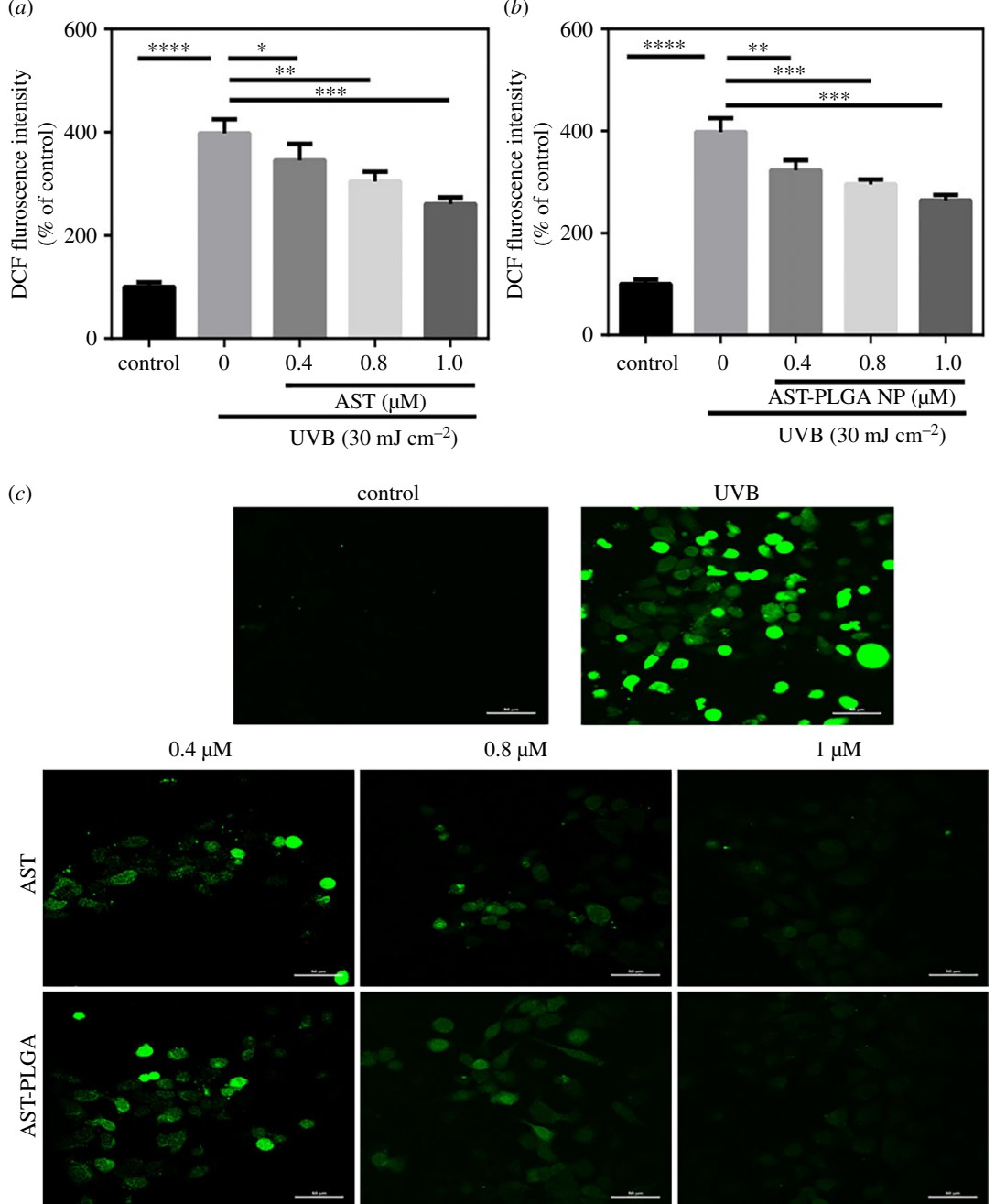

**Figure 6.** Effect of astaxanthin and AST-PLGA NP treatment on UVB-induced ROS generation in HaCaT cells by DCFH-DA assay. The quantitative analysis of intracellular ROS level in HaCaT cells at different concentrations of astaxanthin (*a*) and AST-PLGA NP (*b*). Values were presented as the mean ± s.e.m. ($n = 3$). *$p < 0.05$, **$p < 0.001$, ***$p < 0.005$, ****$p < 0.0001$ compared with UVB- treated cells. (*c*) The fluorescent images of ROS level in cells for different concentrations by laser scanning confocal microscope. Scale bar, 50 μm.

maybe the generation of ROS [48]. The relative $\Delta\Psi$m value was analysed by flow cytometry and the results are shown in figure 7*a,b*. UVB radiation causes reduced red fluorescence and increased green fluorescence, which means $\Delta\Psi$m was low (figure 7*c*). Compared with control cells, the relative $\Delta\Psi$m value of UVB treatment was about twofold higher. Different concentrations of astaxanthin and AST-PLGA NP could enhance the value of $\Delta\Psi$m. Furthermore, the results of the laser scanning confocal microscope for astaxanthin and AST-PLGA NP significantly rescued mitochondrial membrane depolarization and varies with concentrations.

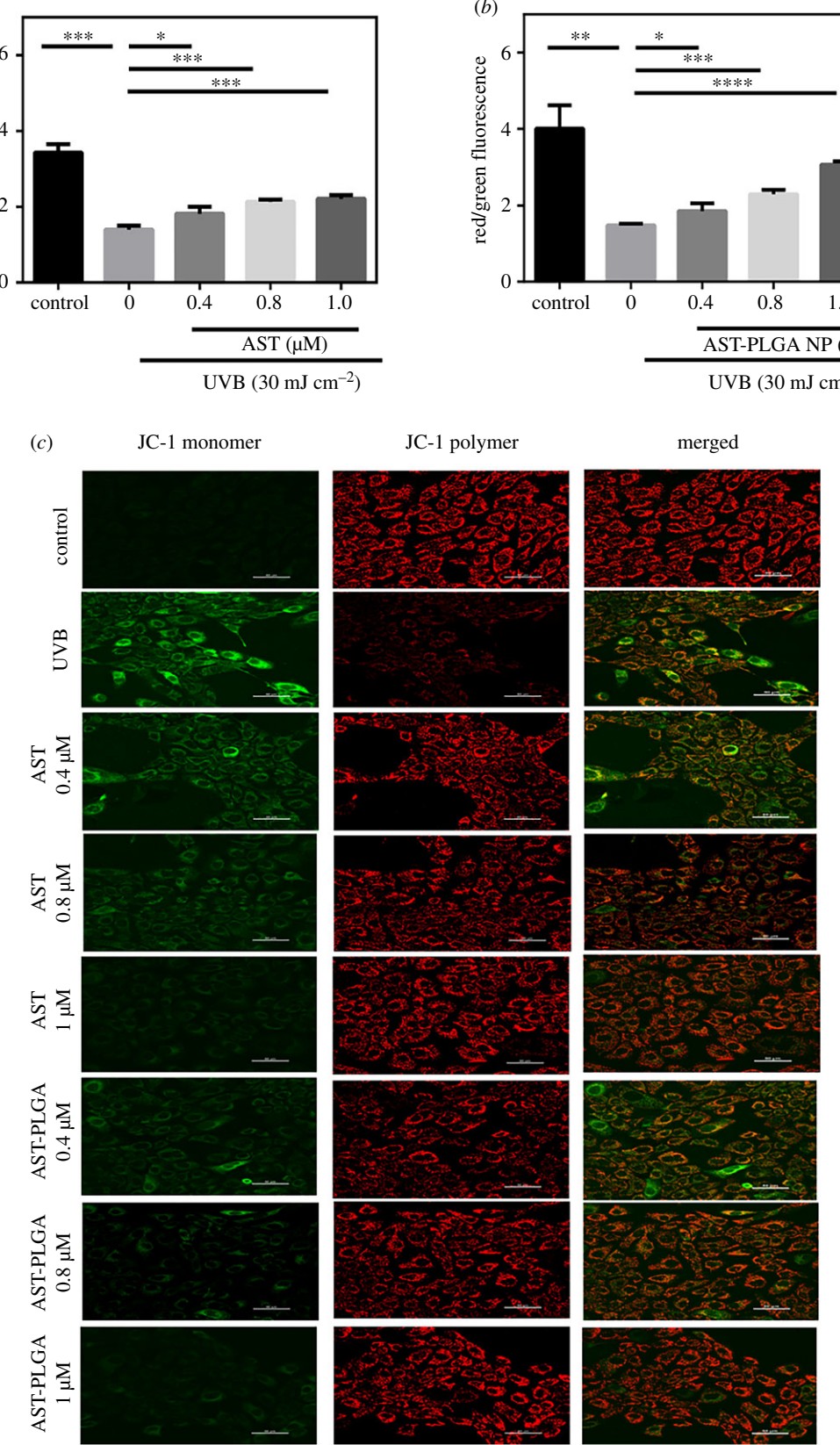

**Figure 7.** Effect of astaxanthin and AST-PLGA NP treatment on UVB-induced mitochondrial membrane depolarization in HaCaT cells by the fluorescent dye JC-1. The quantitative analysis of the relative $\Delta\Psi$m in HaCaT cells at different concentrations of astaxanthin (*a*) and AST-PLGA NP (*b*) by flow cytometry. The results are presented as the mean ± s.e.m. (*n* = 3). *$p < 0.05$; **$p < 0.001$; ***$p < 0.005$; ****$p < 0.0001$; compared with UVB-treated cells. (*c*) The fluorescent images of $\Delta\Psi$m in cells for different concentrations by laser scanning confocal microscope.

# 4. Conclusion

In this research, we have successfully developed the PLGA nano drug delivery system loaded with astaxanthin using the emulsion solvent evaporation technique. The four-factor and three-level BBD optimized the parameters and obtained the optimal process conditions. With this, we synthesized AST-PLGA NP with the concentration of PLGA 10 mg ml$^{-1}$, the concentration of astaxanthin 0.81 mg ml$^{-1}$, water volume 3 ml and sonication time 0.5 min. This synthesized method optimized the AST-PLGA NP with an EE of $96.42 \pm 0.73\%$; DL capacity of $7.19 \pm 0.12\%$ and size of $154.4 \pm 0.35$ nm, indicating the good drug delivery capacity of AST-PLGA NP. In addition, the results of SEM and TEM showed that the nanoparticles were discrete and spherical in shape and displayed a good size distribution. At the same time, FTIR studies confirmed that astaxanthin had been successfully loaded into PLGA NP. XRD and DSC studies demonstrated that astaxanthin existed in the form of dispersed amorphous or disordered crystals in the molecular state while formed a solid solution state in the polymer matrix. PLGA NP were biocompatible and could be taken up by HaCaT cells in a time-dependent manner. In particular, AST-PLGA NP showed better antioxidant activity compared to pure astaxanthin in the UVB radiation photodamage model of HaCaT cells. Furthermore, AST-PLGA NP resists the photodamage in HaCaT cells by reducing ROS levels and restoring mitochondrial membrane potential. Therefore, these results may provide a new means to solve the problem such as high hydrophobicity and poor chemical stability of astaxanthin and demonstrate the therapeutic application of PLGA-encapsulated astaxanthin nanoparticles in skin diseases. All together, the AST-PLGA NP possesses the potential application in the field of cosmetics and might be considered as a potential therapeutic component for skin disease.

Data accessibility. Our data are available from the Dryad Digital Repository: https://doi.org/10.5061/dryad.b8gtht77s [49].
Authors' contributions. F.H. and L.Y. designed the study. F.H. and W.L. performed all experiments. F.H. analysed the data and wrote the rough draft of the manuscript. K.W. and F.K. revised the manuscript. All authors gave final approval for publication.
Competing interests. The authors declare no competing interests.
Funding. Financial support came from the International Cooperation Projects of Guangdong Provincial Science and Technology (grant no. 2015A050502013).
Acknowledgements. This paper was improved thanks to Dattatrya Shetti and Canlong Mo for their technical support.

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
