## [Reviewer comments · Royal Society Open Science]

Review History

RSOS-191184.R0 (Original submission)

Review form: Reviewer 1

Is the manuscript scientifically sound in its present form?

Yes

Are the interpretations and conclusions justified by the results?

Yes

Is the language acceptable?

Yes

Do you have any ethical concerns with this paper?

No

Have you any concerns about statistical analyses in this paper?

No

Recommendation?

Accept with minor revision (please list in comments)

Comments to the Author(s)

I have no corrections to propose, except for a minor concern. I think that the authors have to briefly discuss in the conclusion the clinical potential of their results. This is a recent review on the role of astaxanthin in skin physiology and its clinical implications in dermatology. I think it will be useful to your topic and reference list (Nutrients. 2018 Apr 22;10(4).)

Review form: Reviewer 2**Is the manuscript scientifically sound in its present form?**

Yes

Are the interpretations and conclusions justified by the results?

Yes

Is the language acceptable?

No

Do you have any ethical concerns with this paper?

No

Have you any concerns about statistical analyses in this paper?

No

Recommendation?

Accept with minor revision (please list in comments)

Comments to the Author(s)

In this work, the authors used Box-Behnken experimental design and established appropriate condition for astaxanthin-loaded PLGA nanoparticle preparation. The physicochemical properties and cell uptake studies clearly showed that the nanoparticle possess appropriate properties for a drug carrier system. UVB induced photodamage model was successfully established in Hacat cells and cell viability assay was performed to study the antioxidative properties of the nanoparticle. The astaxanthin encapsulated PLGA nanoparticle secure the photodamage caused by UVB in Hacat cells by decreasing the ROS level. Nanoparticle treatment recovered the Hacat cells from the UVB damaged mitochondrial potential in dose dependent manner. Altogether the study indicated that astaxanthin-loaded PLGA nanoparticle treatment can recover Hacat cells from UVB damage and has potential to use in cosmetic. The novelty of the work is good enough for this Journal , but I still have some comments for improvement.

1) This work has rich content and describes the application and prospect of PLGA-loaded astaxanthin nanoparticles in Hacat UVB damage model. I suggest add some more latest literatures.

2) The experiment is well performed and clearly explained, but I recommend in English language correction.

3) The conclusion part is not appropriately written and need to be improved in detail.

Decision letter (RSOS-191184.R0)

08-Sep-2019

Dear Miss Hu

On behalf of the Editors, I am pleased to inform you that your Manuscript RSOS-191184 entitled "Optimization and Characterization of PLGA nanoparticles loaded with Astaxanthin and evaluation of anti-photodamage effect in vitro" has been accepted for publication in Royal Society Open Science subject to minor revision in accordance with the referee suggestions. Please find the referees' comments at the end of this email.

The reviewers and handling editors have recommended publication, but also suggest some minor revisions to your manuscript. Therefore, I invite you to respond to the comments and revise your manuscript.

- Ethics statement

- Data accessibility

<http://datadryad.org/submit?journalID=RSOS&manu=RSOS-191184>

- Competing interests

- Authors' contributions

- Acknowledgements

- Funding statement

Because the schedule for publication is very tight, it is a condition of publication that you submit the revised version of your manuscript before 17-Sep-2019. Please note that the revision deadline will expire at 00.00am on this date. If you do not think you will be able to meet this date please let me know immediately.

- 1) A text file of the manuscript (tex, txt, rtf, docx or doc), references, tables (including captions) and figure captions. Do not upload a PDF as your "Main Document";
- 2) A separate electronic file of each figure (EPS or print-quality PDF preferred (either format should be produced directly from original creation package), or original software format);
- 3) Included a 100 word media summary of your paper when requested at submission. Please ensure you have entered correct contact details (email, institution and telephone) in your user account;
- 4) Included the raw data to support the claims made in your paper. You can either include your data as electronic supplementary material or upload to a repository and include the relevant doi

within your manuscript. Make sure it is clear in your data accessibility statement how the data can be accessed;

5) All supplementary materials accompanying an accepted article will be treated as in their final form. Note that the Royal Society will neither edit nor typeset supplementary material and it will be hosted as provided. Please ensure that the supplementary material includes the paper details where possible (authors, article title, journal name).

on behalf of Dr Shaked Ashkenazi (Associate Editor)
openscience@royalsociety.org

Reviewer comments to Author:

Reviewer: 1

Comments to the Author(s)

I have no corrections to propose, except for a minor concern. I think that the authors have to briefly discuss in the conclusion the clinical potential of their results. This is a recent review on the role of astaxanthin in skin physiology and its clinical implications in dermatology. I think it will be useful to your topic and reference list (Nutrients. 2018 Apr 22;10(4).)

Reviewer: 2

Comments to the Author(s)

In this work, the authors used Box-Behnken experimental design and established appropriate condition for astaxanthin-loaded PLGA nanoparticle preparation. The physicochemical properties and cell uptake studies clearly showed that the nanoparticle possess appropriate properties for a drug carrier system. UVB induced photodamage model was successfully established in Hacat cells and cell viability assay was performed to study the antioxidative properties of the nanoparticle. The astaxanthin encapsulated PLGA nanoparticle secure the photodamage caused by UVB in Hacat cells by decreasing the ROS level. Nanoparticle treatment recovered the Hacat cells from the UVB damaged mitochondrial potential in dose dependent manner. Altogether the study indicated that astaxanthin-loaded PLGA nanoparticle treatment can recover Hacat cells from UVB damage and has potential to use in cosmetic. The novelty of the work is good enough for this Journal, but I still have some comments for improvement.

- 1) This work has rich content and describes the application and prospect of PLGA-loaded astaxanthin nanoparticles in Hacat UVB damage model. I suggest add some more latest literatures.
- 2) The experiment is well performed and clearly explained, but I recommend in English language correction.
- 3) The conclusion part is not appropriately written and need to be improved in detail.

Author's Response to Decision Letter for (RSOS-191184.R0)

See Appendix A.

Decision letter (RSOS-191184.R1)

25-Sep-2019

Dear Dr Hu,

I am pleased to inform you that your manuscript entitled "Optimization and Characterization of PLGA nanoparticles loaded with Astaxanthin and evaluation of anti-photodamage effect in vitro" is now accepted for publication in Royal Society Open Science.

Royal Society Open Science operates under a continuous publication model (<http://bit.ly/cpFAQ>). Your article will be published straight into the next open issue and this

will be the final version of the paper. As such, it can be cited immediately by other researchers. As the issue version of your paper will be the only version to be published I would advise you to check your proofs thoroughly as changes cannot be made once the paper is published.

Kind regards,

on behalf of Dr Shaked Ashkenazi (Associate Editor) and the Subject Editor
openscience@royalsociety.org

Appendix A

Response to Reviewer

Dear Editors and Reviewers:

Thank you for your letter and for the reviewers' comments concerning our manuscript entitled "Optimization and Characterization of PLGA nanoparticles loaded with Astaxanthin and evaluation of anti-photodamage effect in vitro". Those comments are all valuable and very helpful for revising and improving our paper. And according to the comments, we have revised the manuscript as follows.

Reviewer comments to Author:

Reviewer: 1

Reviewer comment 1: I think that the authors have to briefly discuss in the conclusion the clinical potential of their results.

Author reply: As your great suggestion, we have discussed the clinical potential of AST-PLGA NP in Conclusion—"Therefore, these results may provide a new means to solve the problem such as high hydrophobicity and poor chemical stability of astaxanthin and demonstrate the therapeutic application of PLGA-encapsulated astaxanthin nanoparticles in skin diseases. All together, the AST-PLGA NP possessed the potential application in the field of cosmetics and might be considered as a potential therapeutic component for skin disease."

Reviewer comment 2: This is a recent review on the role of astaxanthin in skin physiology and its clinical implications in dermatology. I think it will be useful to your topic and reference list (Nutrients. 2018 Apr 22;10(4).)

Author reply: Thank you for your great suggestion. We have already added it in Introduction—"The anti-wrinkle and anti-oxidation effects of astaxanthin reflect its various health benefits and important nutritional health applications in dermatology[24]."

Reviewer:2

Reviewer comment 1: This work has rich content and describes the application and prospect of PLGA-loaded astaxanthin nanoparticles in Hacat UVB damage model. I suggest adding some more latest literatures.

Author reply: Thank you for your suggestion and comments. we have added more references in Introduction—"Skin, the largest organ in the human body, plays a major role as the protective barrier against harmful external agents such as ultraviolet (UV) radiation, dehydration, temperature changes, and pathogens[21]. Excessive exposure to UV radiation remains a major risk factor for melanoma and non-melanoma skin cancers, especially exposure to UVB radiation can generate excessive reactive oxygen species in cells, that can induce many deleterious effects, including DNA damage,

oxidative stress, photoaging, inflammation, and carcinogenesis[22, 23]. The anti-wrinkle and anti-oxidation effects of astaxanthin reflect its various health benefits and important nutritional health applications in dermatology[24]. Naoki et al. evaluated the effects of astaxanthin on UV-induced skin degradation in 23 healthy Japanese participants and demonstrated the protective and safe nature of astaxanthin[25]. Moreover, Hung et al. found that barrier defects caused by ultraviolet radiation may increase the skin penetration of polymer nanoparticles[26].”

Reviewer comment 2: The experiment is well performed and clearly explained, but I recommend in English language correction.

Author reply: We are sorry for several typos and grammatical errors. We have carefully corrected all typos and grammatical errors throughout the manuscript.

Reviewer comment 3: The conclusion part is not appropriately written and need to be improved in detail.

Author reply: Thank you for your kind comment. We have revised the conclusions in detail —“In this research, we had successfully developed the PLGA nano drug delivery system loaded with astaxanthin using the emulsion solvent evaporation technique. The four-factor and three-level Box-Behnken design optimized the parameters and obtained the optimal process conditions. With this, we synthesized the AST-PLGA NP with the concentration of PLGA 10 mg/mL, the concentration of astaxanthin 0.81 mg/mL, water volume 3 mL, and sonication time 0.5 min. This synthesized method optimized the AST-PLGA NP with an encapsulation efficiency of $96.42 \pm 0.73\%$; drug loading capacity of $7.19 \pm 0.12\%$ and size of 154.4 ± 0.35 nm, indicating the good drug delivery capacity of AST-PLGA NP. In addition, the results of SEM and TEM showed that the NPs were discrete and spherical in shape, and displayed a good size distribution. At the same time, FT-IR studies confirmed that astaxanthin had been successfully loaded into PLGA nanoparticles. XRD and DSC studies demonstrated that astaxanthin existed in the form of dispersed amorphous or disordered crystals in the molecular state, while forms a solid solution state in the polymer matrix. PLGA nanoparticles were biocompatible and could be taken up by HaCaT cells in a time dependent manner. In particular, AST-PLGA NP showed better antioxidant activity compared to pure astaxanthin in the UVB radiation photodamage model of Hacat cell. Furthermore, AST-PLGA NP resists the photodamage in HaCaT cells by reducing ROS levels and restoring mitochondrial membrane potential. Therefore, these results may provide a new means to solve the problem such as high hydrophobicity and poor chemical stability of astaxanthin and demonstrate the therapeutic application of PLGA-encapsulated astaxanthin nanoparticles in skin diseases. All together, the AST-PLGA NP possessed the potential application in the field of cosmetics and might be considered as a potential therapeutic component for skin disease.”

We tried our best to improve the manuscript and made some changes in the manuscript. These changes will not influence the content and framework of

the paper. And here we did not list the changes but marked in yellow in "Main Document-tracked changes". We appreciate for Editors/Reviewers' warm work earnestly, and hope that the correction will meet with approval.

Best wishes!

Yours sincerely,

Hu Fangbin